# Capillary whole-blood IgG-IgM COVID-19 self-test as a serological screening tool for SARS-CoV-2 infection adapted to the general public

Serge Tonen-Wolyec[1,2], Raphael Dupont[3], Salomon Batina-Agasa[2], Marie-Pierre Hayette[4], Laurent Bélec[5,6]*

**1** Ecole Doctorale Régionale D'Afrique Centrale en Infectiologie Tropicale, Franceville, Gabon, **2** Faculty of Medicine and Pharmacy, University of Kisangani, Kisangani, The Democratic Republic of the Congo, **3** BioSynex, Strasbourg, France, **4** Department of Clinical Microbiology, University Hospital of Liège, Liege, Belgium, **5** Laboratoire de Virologie, Hôpital Européen Georges Pompidou, Assistance Publique-Hôpitaux de Paris, Paris, France, **6** Université Paris Descartes, Sorbonne Paris Cité, Paris, France

* laurent.belec@aphp.fr

**Data Availability Statement:** All relevant data are within the manuscript and its Supporting Information files.

## Abstract

The practicability of a prototype capillary whole-blood IgG-IgM COVID-19 self-test (Exacto® COVID-19 self-test, Biosynex Swiss SA, Freiburg, Switzerland) as a serological screening tool for SARS-CoV-2 infection adapted to the general public was evaluated in a cross-sectional, general adult population study performed between April and May 2020 in Strasbourg, France, consisting of face-to-face, paper-based, semi-structured, and self-administered questionnaires. Practicability was defined as the correct use of the self-test and the correct interpretation of the result. The correct use of self-test was conditioned by the presence of the control band after 15-min of migration. The correct interpretation of the tests was defined by the percent agreement between the tests results read and interpret by the participants compared to the expected results coded by the numbers and verified by trained observers. A total of 167 participants (52.7% female; median age, 35.8 years; 82% with post-graduate level) were enrolled, including 83 and 84 for usability and test results interpretation substudies, respectively. All participants (100%; 95% CI: 95.6–100) correctly used the self-test. However, 12 (14.5%; 95% CI: 8.5–23.6) asked for verbal help. The percent agreement between the tests results read and interpret by the participants compared to the expected results was 98.5% (95% CI: 96.5–99.4). However, misinterpretation occurred in only 2.3% of positive and 1.2% of invalid test results. Finally, all (100%) participants found that performing the COVID-19 self-test was easy; and 98.8% found the interpretation of the self-test results easy. Taken together, these pilot observations demonstrated for the first-time, high practicability and satisfaction of COVID-19 self-testing for serological IgG and IgM immune status, indicating its potential for use by the general public to complete the arsenal of available SARS-CoV-2 serological assays in the urgent context of the COVID-19 epidemic.

**Funding:** This work was partly supported by Biosynex SA. The funders played a role in providing the prototype SARS-CoV-2 test for self-test (Exacto® COVID-19 self-test, Biosynex Swiss SA) and data collection. The study design, analysis, decision to publish, and preparation of the manuscript were not sponsored. Biosynex SA also provided support for this study in the form of salary for Dr. Raphael Dupont. The specific role of this author is articulated in the 'author contributions' section. Dr. Serge Tonen-Wolyec was recipient of ERASMUS+ program between the University of Kisangani, Democratic Republic of the Congo, and the University of Liège, Belgium. There was no additional external funding received for this study.

**Competing interests:** The authors have read the journal's policy and have the following competing interests: Dr. Raphael Dupont is a paid employee of Biosynex SA. The authors would like to declare the following patents/patent applications associated with this research: https://bases-marques.inpi.fr/Typo3_INPI_Marques/ajoutListe?page=1&idObjet=1484785_202032_tmint&scroll=462.4761962890625. This does not alter our adherence to PLOS ONE policies on sharing data and materials.

## Introduction

Severe acute respiratory syndrome coronavirus 2 (SARS-CoV-2), a novel coronavirus that causes Coronavirus Disease 2019 (COVID-19), started in the Wuhan province, China, in December 2019, and was declared by the World Health Organization (WHO) as global pandemic on March 11, 2020 [1–4]. Controlling the outbreak in the community and in hospitals mainly relied on the availability of highly sensitive and specific nucleic acid amplification-based molecular testing for SARS-CoV-2 [5, 6]. Furthermore, it was demonstrated that serological testing looking for specific SARS-CoV-2 IgG and/or IgM may be useful for confirming the diagnosis and care of COVID-19 patients [7–9]. On March 2, 2020, the WHO recommended serological testing in addition of molecular diagnosis, for investigating on-going outbreaks as well as for the diagnosis of strongly suspected patients of SARS-CoV-2 infection with negative RT-PCR [10]. Furthermore, antibody tests for SARS-CoV-2 may constitute one of the keys to fight the SARS-CoV-2 epidemic, in particular, to overcome the period after lockdown [9]. Seropositivity to SARS-CoV-2 antigens would also allow to identify previously infected individuals, including asymptomatic patients [9].

Recently, rapid lateral flow assays for IgG and IgM antibodies produced during the COVID-19 epidemic have been developed [11]. Several reports have shown that COVID-19 IgG/IgM lateral flow immunoassays may be a reliable tool to diagnose SARS-CoV-2 infection from 14 days of onset of symptoms [12, 13]. In some countries, rapid diagnostic testing for COVID-19 has been incorporated into the local guidelines for testing asymptomatic contacts of positive cases, at day 14 of home surveillance [14]. These easy to use IgG-IgM combined tests allow rapid screening with capillary blood samples. The tests are simple, qualitative, visually interpretable, and give a result within 10 to 15 minutes. A positive serology allows to determine whether a person has already been infected by SARS-CoV-2. Serologic tests will be needed to assess the response to vaccine candidates and to map levels of immunity in communities. These rapid tests could be particularly interesting for developing countries for testing patients at the bedside or any other locations where laboratory facilities are lacking.

HIV self-testing constitutes a novel innovative approach to make testing more accessible, confidential, and available at non-traditional venues, such as pharmacies and community venues, as well as in the home, as it offers a discreet, convenient, and empowering way to test [15, 16]. HIV self-testing has demonstrated high acceptability with very convenient usability in various adolescent and adult populations from developed as well as resources-constrained settings [17–21].

To our knowledge, there is no currently reported experience in the literature about self-testing for SRAS-CoV-2 infection. Based on our own experience of HIV self-testing evaluation, we herein aimed at evaluating the practicability of a prototype capillary whole-blood IgG-IgM COVID-19 self-test as a serological screening tool for SARS-CoV-2 infection adapted to the general public.

## Material and methods

### Prototype SARS-CoV-2 test for self-testing

The prototype capillary whole-blood IgG/IgM SARS-CoV-2 self-test (Exacto® COVID-19 self-test, Biosynex Swiss SA, Freiburg, Switzerland) was adapted from the CE IVD-labeled fingerstick whole-blood rapid diagnostic test for IgG and IgM antibodies against SARS-CoV-2 detection (BIOSYNEX® COVID-19 BSS [IgG/IgM], Biosynex Swiss SA), by re-packaging for individual use with the addition of seven components placed in a pouch containing the test cassette, diluent vial, pipette, alcohol wipe, compress, lancet and dressing. The Exacto® COVID-19 self-

test (Biosynex Swiss SA) consists of visually read, qualitative, *in vitro* lateral flow immunoassays for the detection of IgG and IgM antibodies to SARS-CoV-2 in human whole blood, serum, or plasma as an aid in the diagnosis of SARS-COV-2 infection. The targeted protein is the receptor-binding domain (RBD) of the spike surface protein of SARS-CoV-2. During testing, the specimen reacts with SARS-CoV-2 antigen-coated particles in the test cassette. The mixture then migrates upward on the membrane chromatographically by capillary action and reacts with the anti-human IgG in the IgG test line region or/and with the anti-human IgM in the IgM line region. The quantity of blood needed to perform the test is 10 μL.

The analytical performances of the BIOSYNEX® COVID-19 BSS (IgG/IgM) (Biosynex Swiss SA) were evaluated during the COVID-19 epidemic in *Grand Hôpital de l'Est francilien*, Jossigny, France, using two serum sample panels obtained from patients with COVID-19 confirmed by positive nucleic acid amplification-based diagnosis at least 14 days after symptoms onset and from patients randomly selected for whom serum samples were collected before the COVID-19 epidemic (from October 1 to November 30, 2019) (instructions for use 2020). The BIOSYNEX® COVID-19 BSS (IgG/IgM) (Biosynex Swiss SA) showed sensitivity of 97.4% and specificity of 100%, demonstrating high analytical performances allowing convenient management of suspected on-going and past-infections. Furthermore, this rapid diagnostic test is recommended for both SARS-CoV-2-specific IgG and IgM detection by the French Ministry of Health [22], following an official report from the National Reference Center for Respiratory Viruses [*Centre National de Référence Virus des infection respiratoires (dont la grippe)*], Institut Pasteur, Paris, because the test fulfilled the criteria of the minimal analytical performances [*i.e.* minimum sensitivity of 90% (or even 95%) and minimum specificity of 98%] of serological tests detecting the antibodies directed against SARS-CoV-2, defined on April 16, 2020 by the so-called *Haute Autorité de Santé* [23]. The simplified instructions for use of the Exacto® COVID-19 self-test (Biosynex Swiss SA) comprised an easy-to-read leaflet in French and English, in A3 format color printing. As an example, the paper-based and video-based instructions for use are depicted as S1 Appendix and S1 Video.

## Study design and recruitment of participants

The practicability evaluation of the Exacto® COVID-19 self-test (Biosynex Swiss SA) was a cross-sectional study, consisting of face-to-face, paper-based, semi-structured, and self-administrated questionnaires. This survey was performed between April and May 2020 by home-based recruitment of adult volunteers using a door-to-door community approach, in 15 neighborhoods of Strasbourg and its suburbs, France. Due to the limited movement during the confinement period in France, especially in the province of Alsace (now "Grand Est") for which Strasbourg is the capital city, the choice of these neighborhoods and its suburbs was based on their easy accessibility and their high prevalence of reported cases of SARS-CoV-2 infection [24].

All participants accepted voluntarily to be included. Eligible participants had an age ≥ 18 years, wanted to know their SRAS-CoV-2 serology status, were capable to speak and read in French, and gave their consent to participate in the study. All trained individuals (physicians, nurses, and biologist) in rapid diagnostic tests were excluded. Informed written consent was signed by all participants. Ethical approval for this study was obtained from the local scientific committee of Parc de l'Innovation, Strasbourg, France.

## Practicability study outcomes

The practicability evaluation was divided into four substudies carried out by trained health care professionals, based on previously acquired experience from WHO recommendations for evaluating the practicability of HIV self-tests [17, 18, 25]. Indeed, the practicability was defined

as the correct use of the self-test and the correct interpretation of the result. The correct use of self-test was conditioned by the presence of the control band after 15-min of migration. The correct interpretation of the tests was defined by the percent agreement between the tests results read and interpret by the participants compared to the expected results coded by the numbers and verified by trained observers. As depicted in the Fig 1, all participants were included in substudy 1 concerning the understanding of labeling, while they were randomized into two groups for substudy 2 concerning manipulation of the test and substudy 3 evaluating the interpretation of COVID-19 self-test results, using block randomization of 4. Participants in sub-study 4 were each drawn from the satisfaction questionnaires for substudies 2 and 3.

## Data collection and procedures

Paper-based, self-administered, and structured questionnaires were used to obtain the data on the socio-demographic characteristics, medical history of study participants, participants' understanding of the instructions for use, and participants' opinions or levels of satisfaction about the practicability of the Exacto® COVID-19 self-test (Biosynex Swiss SA). All data related to the observation of manipulation and the interpretation of test results were recorded on the standardized sheets by the observers.

**Substudy 1. Comprehension of labeling.** After receiving a brief explanation of the objectives and conduct of the study, the participants were asked to sign the informed consent form. In a private setting, the participants had the choice between a paper-based instruction for use and a video-based instruction for use, which they were asked to read or watch and understand independently. After their self-declaration of having understood the instruction for use, the participants were asked to fill a questionnaire to gauge their comprehension. To this end, 10 questions restating the key information with closed answers (true, false, or don't know) were asked by the observer on the followings items: 1. Identification of each component of the kit; 2. Manipulation of blood sampling device; 3. Diluent deposit; 4. Possession of a timer; 5. Interpretation of a positive test result; 6. Interpretation of a negative test result; 7. Diagnosis of an invalid test result; 8. Reliability of self-test result; 9. Meaning of a positive result; and 10. Detection of the virus. The participants who correctly answered all 10 questions were considered to have correctly understood the instructions for use.

After this survey, participants were randomized in two groups for evaluation on performing the self-test and the interpretation of test results. In order to achieve this, a sealed randomization envelope was used sequentially. In each group, before starting the survey, a pre-test satisfaction questionnaire was completed by the participants.

**Substudy 2. Observation of manipulation.** In a private setting supervised by an observer, each participant received a box containing the Exacto® COVID-19 self-test (Biosynex Swiss SA). Participants were then asked to carry out the self-test by themselves in front of a trained observer. The observer was responsible for recording the respect or not of each step, provide verbal assistance (mimicking telephone support), difficulty, and errors on a standardized sheet. The successful performance of the SARS-CoV-2 self-test was conditioned by the presence of the control band on test strip, and the test results were read and recorded independently by both the participants and the observers. Note that, all individuals with a positive serological result were referred to the laboratory for diagnostic confirmation and to the hospital for management.

**Substudy 3. Interpretation of test results.** In a private setting supervised by an observer, eight standardized test results including four positive tests (one weak positive for IgM, one clearly positive for IgM, one clearly positive for IgG but weak positive for IgM, and one clearly positive for IgM and IgG), two negative tests, and two invalid tests were provided to the

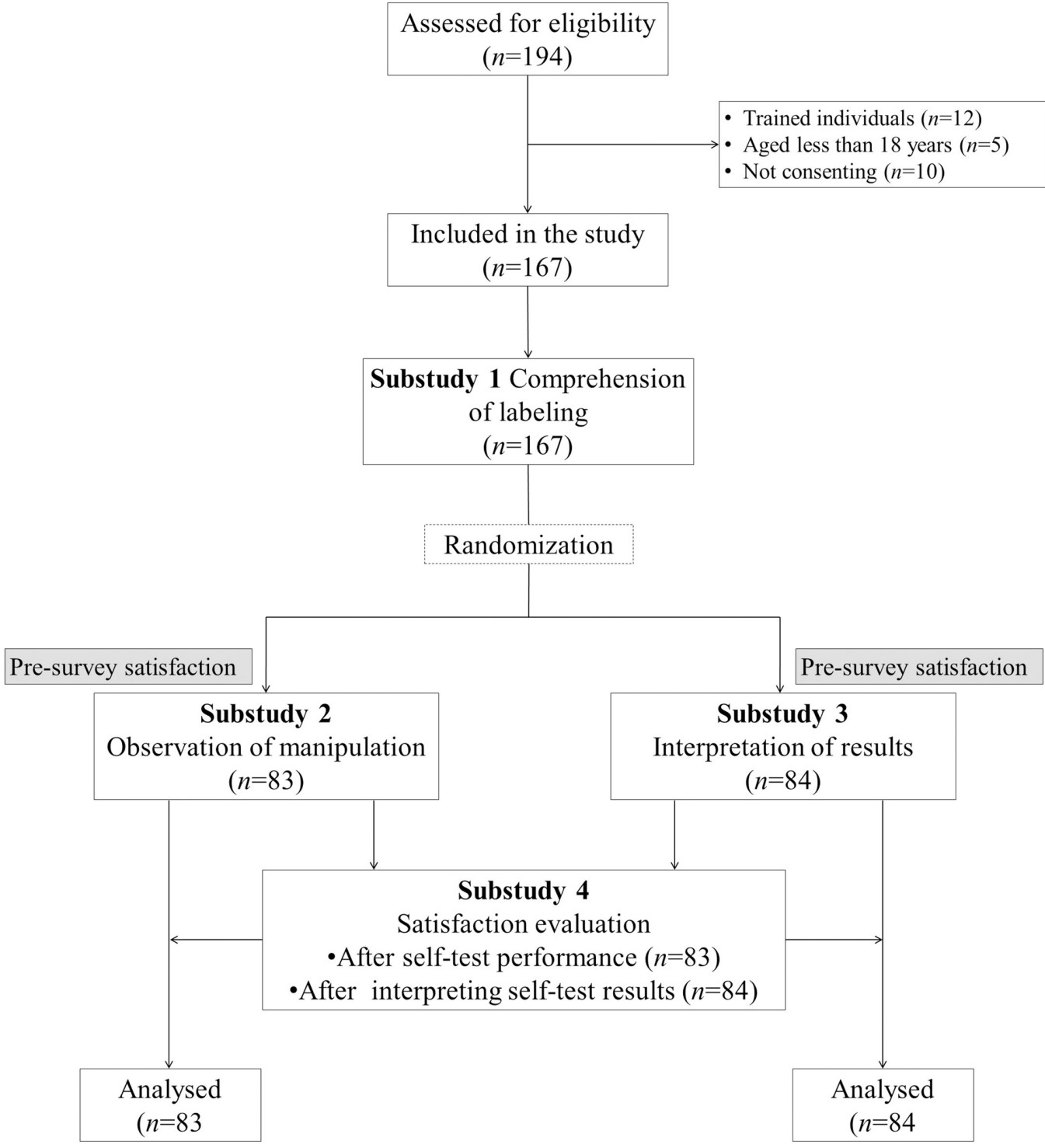

**Fig 1. Flow chart showing the recruitment of study participants, their randomization, and affiliation for each substudy.**

participants for interpretation after random selection of four tests (Fig 2). These standardized tests were coded by numbers to determine the expected results.

### A. Interpretation of test results

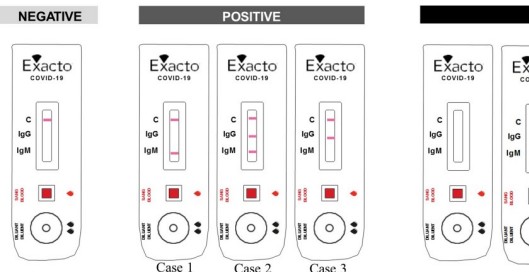

### B. Panel of eight standardized tests

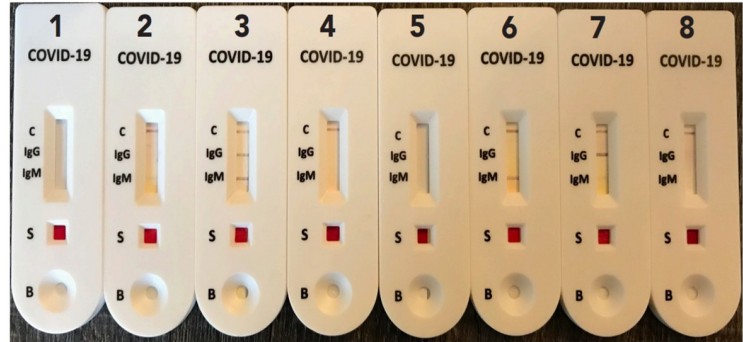

**Fig 2. Interpretation of self-test results. A.** The self-test result was interpreted as negative when a Control line (C) was present and readable and the "IgG" and "IgM" lines were absent. It was positive when a "C" and "IgM" (clearly or poorly readable) (case 1), or "C" and "IgG", or "C", "IgM" (clearly or poorly readable), and "IgG" lines were present. Finally, it was invalid when the "C" line was absent regardless of the presence or absence of the "IgG" and/or "IgM" line. **B.** Panel of 8 Exacto® COVID-19 self-test (Biosynex Swiss SA) cassettes, including 4 positive tests (#2, #3, #6 and #7), 2 negative tests (#4 and #8) and 2 invalid tests (#1 and #5). The #2 and #37 are weakly positive for IgM. Each volunteer randomly drew 4 tests among a panel of 8 and interpreted them with the help of the reading and interpretation scale. The observer noted the number of the drawn test and the result given by the participant.

**Substudy 4. Satisfaction questionnaire.** Finally, the participants fulfilled the satisfaction questionnaire concerning their experiences with the COVID-19 self-test including understanding of instructions for use, the identification of the different components of the kit, the sample collection and transfer, the overall performance of the self-test, the reading and interpretation of test results, and the ability to overcome the difficulties encountered.

## Statistical analysis

All data were entered into an Excel file and analyzed on SPSS 20.0 (Chicago, IL). Descriptive statistics were computed using mean (standard deviation) or median (interquartile range) for normal or skewed distribution, respectively, then, proportions of all categorical variables were calculated for qualitative data. The labeling index for understanding and usability index were defined as the mean of the correct answers for each question related to the understanding of instructions for use and performing of the COVID-19 self-test, respectively. The Wilson score bounds were used to estimate the 95% confidence intervals (CI). Cohen's κ coefficient estimated the concordance between the results read by participants in connection with the expected results [26]. The degree of agreement was determined as ranked by Landlis and Koch [27]. The comparison of data from the post-test satisfaction questionnaire paired to those from the pre-test satisfaction questionnaire was performed by using Mac Nemar's chi-squared pairing test.

# Results

## Study population

A total of 194 individuals were assessed for eligibility, but 27 were excluded because they were trained (n = 12), less than 18 years old (n = 5), or not consenting (n = 10). Finally, 167 were successfully enrolled in the study (substudies 1 and 4), and among them, 83 were assigned after randomization in substudy 2 and 84 in substudy 3 (Fig 1). The demographic characteristics and medical history of study participants are shown in Table 1. Overall, 88 (52.7%) were female. The mean age was 38.6 (SD: 13.8) years, and around one half of participants were aged between 18 and 39 years. The majority (82.0%) of participants had post-graduate education level. The majority (59.3%) had reported no symptoms of COVID-19 in the past two months. Approximately one fifth of participants had previously been screened for SARS-CoV-2 infection by molecular testing of nasopharyngeal swab, of whom 64.7% had a positive result (Table 1).

**Substudy 1.** This substudy evaluated the ability of the 167 study participants to understand the instructions for use of the Exacto® COVID-19 self-test (Biosynex Swiss SA). A large majority (*n* = 155; 92.8%) of participants preferred to use the paper-based instructions whereas only 12 (7.2%) participants used the video-based instructions. The analytical results of the evaluation questionnaire are shown in Table 2. Overall, 149 (89.2%; 95% CI: 83.6–93.1) participants correctly understood the instructions for use, thus correctly answering all 10 questions.

**Table 1. The demographic characteristics and medical history of the 167 study participants.**

| Variable | Items | Number (%) |
|---|---|---|
| **Sex** | | |
| | Male | 79 (47.3) |
| | Female | 88 (52.7) |
| **Age (years)** | | |
| | 18–39 | 88 (52.7) |
| | ≥ 40 | 79 (47.3) |
| Mean (SD) | | 38.6 (13.8) |
| **Educational level** | | |
| | College level | 14 (8.4) |
| | High school level | 16 (9.6) |
| | Post-graduate level | 137 (82.0) |
| **Had the symptoms of COVID-19 in the past two months[#]** | | |
| | Yes | 68 (40.7) |
| | No | 99 (59.3) |
| **Previous COVID-19 molecular testing (nasopharyngeal swab)** | | |
| | Yes | 34 (20.4) |
| | No | 133 (79.6) |
| **Previously diagnosed COVID-19 positive among those previously COVID-19 tested, *n* = 34** | | |
| | Yes | 22 (64.7) |
| | No | 12 (35.3) |

[#] Participants who reported having at least one of the following major symptoms associated or not with minor symptoms were considered to have the COVID-19 symptom: fever, fatigue, dry cough, anosmia and dyspnea. Minor symptoms were: pain, nasal congestion, runny nose, sore throat or diarrhea.

COVID-19: Coronavirus disease 2019; RT-PCR: Reverse transcription-polymerase chain reaction; SD: Standard deviation.

**Table 2. Analytical results of the evaluation questionnaire concerning the ability of the 167 study participants to understand the instruction for use of the Exacto®
COVID-19 self-test (Biosynex Swiss SA) (substudy 1).** The questions raising specific issues concerning the manipulation of the kit, the interpretation of test results, and
the consequence of test results, were asked by the observer and the answers were closed.

| Comprehension of labeling checklist* | Participants' responses | | |
|---|---|---|---|
| | **True** | **False** | **Don't know** |
| | [*number (%)*] | [*number (%)*] | [*number (%)*] |
| **Q1**: "A capital letter is associated with each component of the kit to better identify it during the performance of self-test" | 166 (99.4) | - | 1 (0.6) |
| **Q2**: "The blood collection device (lancet) helps to collect the blood and transfer it immediately into the SQUARE well of self-test with the pipette" | 165 (98.8) | 1 (0.6) | 1 (0.6) |
| **Q3**: "Two drops of diluent should be placed in the same well as the drop of blood" | 2 (1.2) | 163 (97.6) | 2 (1.2) |
| **Q4**: "A timer (watch or mobile) to clock 10 minutes before reading the result is need" | 167 (100) | - | - |
| **Q5**: "Presence of a readable strip next to IgM and/or IgG on the self-test cassette means that the test is positive" | 166 (99.4) | 1 (0.6) | - |
| **Q6**: "Lack of band by test results is interpreted as a negative test" | 4 (2.4) | 162 (97.0) | 1 (0.6) |
| **Q7**: "Lack of control band by test results should be interpreted as an invalid test" | 167 (100) | - | - |
| **Q8**: "Having symptoms less than 10 days before the test does not provide a reliable result" | 157 (94.0) | 7 (4.2) | 3 (1.8) |
| **Q9**: "If the test is positive it means that they have been in contact with the virus" | 163 (97.6) | 3 (1.8) | 1 (0.6) |
| **Q10**: "The Exacto® COVID-19 self-test does not detect the presence of the virus" | 148 (88.6) | 17 (10.2) | 2 (1.2) |
| *Labeling index for understanding (% [95% CI])£* | 97.1 [93.3–98.8] | | |
| *Correct understanding of the instruction for use (n; % [95% CI])#* | 149; 89.2 [83.6–93.1] | | |

* Overall, 155 (92.8%) participants preferred to use the paper-based instruction whereas only 12 (7.2%) participants used the video-based instruction;

£ The labeling index for understanding was defined as the mean of the correct answers for each question;

# The participants who correctly answered all 10 questions were considered to have correctly understood the instructions for use.

CI: Confidence interval; COVID-19: Coronavirus disease 2019; Q: Question.

The labeling index for understanding measuring the mean of the correct answers for each
question was 97.1% (95% CI: 93.3–98.8). The question (Q10) concerning the non-detection of
the virus (SARS-CoV-2) by the self-test showed the highest rate (10.2%) of incorrect response.

**Substudy 2.** This substudy evaluated the ability of participants to use the Exacto®
COVID-19 self-test (Biosynex Swiss SA) in a supervised setting. The results of the question-
naire are shown in Table 3. Overall, all participants (100%; 95% CI: 95.6–100) performed the
self-test and succeeded in obtaining a valid test result with an overall usability index of 98.5%
(95% CI: 93.0–99.7). Seventy (83.1%; 95% CI: 75.0–90.6) participants correctly used the self-
test without any difficulties, errors, and help, whereas 12 (14.5%; 95% CI: 8.5–23.6) had asked
for verbal help. The identification of the different components of the kit, the use of the lancet
and pipette, and the transfer of blood were the steps requiring the most frequent verbal help in
1.2%, 2.4%, 8.4%, and 2.4%, respectively (Table 3). Interestingly, all participants (n = 6; 7.2%)
using the video instructions performed the self-test easily (usability index of 100%) without
any difficulties, errors, and help. Overall, the mean time of self-test performance (since the
opening of the box until the migration step) was 8.8 (SD: 3.0) minutes. Note that, in this sub-
study, 11 (13.3%) people had a positive results with the self-test, and they were referred to a
clinically certified laboratory for result confirmation.

**Substudy 3.** This substudy evaluated the ability of participants to read and interpret the
COVID-19 self-test results after random selection of four tests from a panel of eight standardized
tests. The results are depicted in Fig 3. Overall, 336 standardized tests were read and interpreted
by the 84 participants, including 171 positive, 84 negative, and 81 invalid test results. A total of
331 (98.5%; 95% CI: 96.5–99.4) tests were correctly interpreted, whereas 5 (1.5%; 95% CI: 0.6–
3.5) tests were misinterpreted. Misinterpretation occurred in 2.3% (n = 4) of positive tests (all
tests were weakly positive for IgM tests falsely interpreted as negative) and in 1.2% (n = 1) of

**Table 3. Analytical results of the manipulation observation concerning the ability of the randomly selected 83 study participants to correctly use each step of the Exacto® COVID-19 self-test (Biosynex Swiss SA) autonomously or with verbal help (substudy 2).**

| Usability checklist* | Successful manipulation | | Need for verbal help |
|---|---|---|---|
| | Yes | No | Yes |
| | [*number (%)*] | [*number (%)*] | [*number (%)*] |
| 1. Did the participant read the instruction for use? | 83 (100) | - | - |
| 2. Did the participant easily identify the different components of the kit? | 82 (98.8) | 1 (1.2) | 1 (1.2) |
| 3. Did the participant wash his hands? | 83 (100) | - | - |
| 4. Did the participant properly remove the test cassette from the aluminum pouch? | 81 (97.6) | 2 (2.4) | - |
| 5. Did the participant open the diluent vial correctly? | 83 (100) | - | - |
| 6. Did the participant disinfect his finger correctly? | 83 (100) | - | - |
| 7. Did the participant wipe residual alcohol with the compress? | 82 (98.8) | 1 (1.2) | - |
| 8. Did the participant have difficulty lancing their finger? | 2 (2.4) | 81 (97.6) | 2 (2.4) |
| 9. Did the participant have difficulty forming a blood droplet? | 1 (1.2) | 82 (98.8) | - |
| 10. Did the participant have difficulty using the pipette correctly until it was filled up to the blank line? | 7 (8.4) | 76 (91.6) | 7 (8.4) |
| 11. Did the participant correctly transfer and deposit the blood into the SQUARE well of the test cassette? | 81 (97.6) | 2 (2.4) | 2 (2.4) |
| 12. Did the participant shed two drops of diluent in the ROUND well of the test cassette? | 83 (100) | - | - |
| 13. Did the Participant obtain an interpretable result at the end of the process despite a missed or incorrect step?# | 83 (100) | - | - |
| *Usability index and overall need for help (% [95% CI])£* | 98.5 [93.0–99.7] | | 14.5 [8.5–23.6] |
| *Correct use without difficulties, errors, and helps (n; % [95% CI])* | 70; 83.1 [75.0–90.6] | | |
| *Average time of manipulation (minutes [SD])* | 8.8 [3.0] | | |

* 6 (7.2) participants had used the video-based instruction for use; among them the usability index was estimated to 100% without any difficulties, errors, and help;

# The result was considered interpretable when a control strip was readable after the migration time recommended by the manufacturer; in the present series, 11 (13.3%) participants had a positive self-test result;

£ The usability index was defined as the mean of the correct answers for each question.

CI: Confidence interval; SD: Standard deviation.

invalid tests falsely interpreted as negative. Cohen's κ coefficient between the results of reading by participants and the expected results was 0.98, demonstrating an excellent concordance.

**Substudy 4.** This substudy assessed the pre-test and post-test satisfaction of participants concerning the instructions for use (substudy 1), performing the COVID-19 self-test (substudy 2), and the interpretation of test results (substudy 3). The results of the questionnaire are shown in Table 4. The understanding of the instructions for use of the self-test was considered easy in pre-test satisfaction questionnaire as well as in post-test period (100% *versus* 97.6%; not significant). However, 92.8% of participants found that the sample collection was very easy in pre-test satisfaction questionnaire whereas this satisfaction decreased after self-testing to 71.1%, yielding a difference of -21,7 (95% CI: -31.7 to -11.7; $P<0.001$). Similar decrease was observed with the satisfaction of sample transfer (81.2% *versus* 60.2%; difference: -21.0% [95% CI: -30.9 to 11.1]; $P<0.001$). Concerning the interpretation of test results, the participants found it easy in pre-test satisfaction questionnaire as well as in post-test period (100% *versus* 98.8%; not significant). Finally, when asked about the ability to surmount the difficulties encountered during COVID-19 self-testing, all (100%) participants found it easy (97.0% very easy; 3.0% rather easy).

## Discussion

We herein report on our recent experience during the last COVID-19 epidemic peak period of the practicability of a prototype capillary whole-blood COVID-19 self-test for IgG and IgM

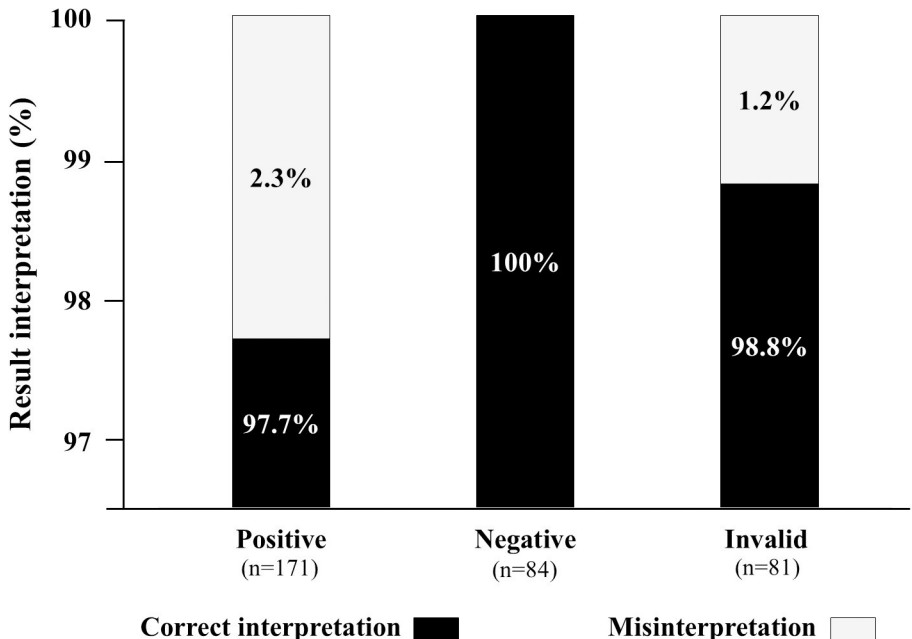

**Fig 3. Stacked columns showing the ability of participants to read and interpret (correctly or incorrectly) the 336 results of the Exacto® COVID-19 self-test (Biosynex Swiss SA) obtained from random selection of a panel of 8 standardized tests, including four positive, two negative, and two invalid test results.**

against SARS-CoV-2 serological screening among adult volunteers living in France. Our assessment of usability was made with reference to our previous experience in evaluating HIV self-testing according to the WHO recommendations [25]. Overall, the vast majority of participants correctly understood the instructions for use, showed good ability to carry out the self-testing procedure in order to obtain a valid test result, and demonstrated to be capable to correctly interpret the test results with high degree of satisfaction. Only a minority of participants needed verbal help, and only 1.5% of test results were misinterpreted. Taken together, our pilot study generated for the first-time to our knowledge evidence on generally good practicability of COVID-19 self-testing for serological IgG and IgM immune status, despite some limitations. These findings also provide the observational basis for the possibility of using with high confidence self-tests harboring 3 bands of interest, *i.e.* in the case of the prototype COVID-19 self-test, the control, IgG and IgM bands. Finally, our observations lay the foundations for the potential large-scale use of COVID-19 self-test in lay adults, at least Europeans of high educational attainment, to complete the arsenal of available serological tests used to assess the immune status vis-a-vis SARS-CoV-2.

## Substudy 1

The learning process in different fields of science needs to link theory to practice [28]. The expected results of substudy 1 are, therefore, important for the following practicability substudies 2 and 3, because it is mandatory to check that the instructions for use can be read and understood by all users. Our findings showed that 89.2% of participants correctly answered all 10 questions indicating generally correct understanding of the key messages delivered by the instructions for use of the Exacto® COVID-19 self-test (Biosynex Swiss SA), with an overall rate of good responses of 97.1%. These satisfactory results may be explained in part by the high post-graduate education level of the majority of study participants. Indeed, previous

**Table 4. Items and results of the pre-test and post-test satisfaction questionnaire and concerning the instruction notice (substudy 1), the performing of the Exacto COVID-19 self-test (Biosynex Swiss SA) (substudy 2), and the interpretation of test results (substudy 3).**

| Satisfaction questionnaire | Pre-test satisfaction | Post-test satisfaction | Difference* | | P-value# |
|---|---|---|---|---|---|
| | | | % [95% CI] | | |
| | [*number (%)*] | [*number (%)*] | | | |
| How did you find the understandability of instructions for use of self-test? (N = 167) | | | | | |
| Very easy | 156 (93.4) | 153 (91.6) | -1.8 (-5.1 to +1.5) | | NS |
| Rather easy | 11 (6.6) | 10 (6.0) | -0.6 (-3.3 to +2.1) | | NS |
| Rather difficult | 0 (0) | 2 (1.2) | +1.2 (-1.8 to +4.2) | | NS |
| Very difficult | 0 (0) | 2 (1.2) | +1.2 (-1.8 to +4.2) | | NS |
| How did you find the identification of the different components of the self-test kits? (N = 83) | | | | | |
| Very easy | 81 (97.6) | 80 (96.4) | -1.2 (-6.5 to +4.3) | | NS |
| Rather easy | 2 (2.4) | 3 (3.6) | +1.2 (-4.1 to +6.5) | | NS |
| Rather difficult | 0 (0) | 0 (0) | - | | NA |
| Very difficult | 0 (0) | 0 (0) | - | | NA |
| How did you find the sample collection? (N = 83) | | | | | |
| Very easy | 77 (92.8) | 59 (71.1) | -21.7 (-31.7 to -11.7) | | <0.001 |
| Rather easy | 5 (6.0) | 20 (24.1) | +18.1 (+11.3 to +27.7) | | <0.001 |
| Rather difficult | 0 (0) | 1 (1.2) | +1.2 (-4.1 to +6.5) | | NS |
| Very difficult | 1 (1.2) | 3 (3.6) | +2.4 (-3.5 to 8.3) | | NS |
| How did you find the sample transfer? (N = 83) | | | | | |
| Very easy | 68 (81.2) | 50 (60.2) | -21.0 (-30.9 to 11.1) | | <0.001 |
| Rather easy | 14 (16.9) | 25 (30.1) | +13.2 (+4.3 to +22.1) | | 0.043 |
| Rather difficult | 0 (0) | 2 (2.4) | +2.4 (-3.5 to 8.3) | | NS |
| Very difficult | 1 (1.2) | 6 (7.2) | +6.0 (-1.3 to +13.3) | | NS |
| How did you find the overall performance of self-test? (N = 83) | | | | | |
| Very easy | 80 (96.4) | 77 (92.8) | -3.6 (-10.1 to +2.9) | | NS |
| Rather easy | 2 (2.4) | 6 (7.2) | +4.8 (-2.1 to +11.7) | | NS |
| Rather difficult | 1 (1.2) | 0 (0) | -1.2 (-6.5 to +4.3) | | NS |
| Very difficult | 0 (0) | 0 (0) | - | | NA |
| How did you find the reading of strips after migration? (N = 84) | | | | | |
| Very easy | 73 (86.9) | 70 (83.3) | -3.6 (-10.0 to +3.0) | | NS |
| Rather easy | 8 (9.5) | 10 (11.9) | +2.4 (-3.4 to 8.4) | | NS |
| Rather difficult | 2 (2.4) | 3 (3.6) | +1.2 (-4.0 to +6.4) | | NS |
| Very difficult | 1 (1.2) | 1 (1.2) | - | | NA |
| How did you find the interpretation of self-test results? (N = 84) | | | | | |
| Very easy | 76 (90.5) | 76 (90.5) | - | | NA |
| Rather easy | 8 (9.5) | 7 (8.3) | -1.2 (-6.4 to +4.2) | | NS |
| Rather difficult | 0 (0) | 0 (0) | - | | NA |
| Very difficult | 0 (0) | 1 (1.2) | +1.2 (-4.0 to +6.4) | | NS |
| How did you find your ability to surmount the difficulties encountered? (N = 167) | | | | | |
| Very easy | - | 162 (97.0) | NA | | NA |
| Rather easy | - | 5 (3.0) | NA | | NA |
| Rather difficult | - | 0 | NA | | NA |
| Very difficult | - | 0 | NA | | NA |

* Difference and CI were assessed with the Wilson score bounds using data collected in the post-test satisfaction questionnaire paired to those from the pre-test satisfaction questionnaire;

# P-value calculated using Mac Nemar's test of paired data.

CI: Confidence interval; NA: Not applicable; NS: Not significant.

experience from HIV self-testing showed that insufficient educational level constitutes a great challenge in the comprehension of the instructions for use [18, 29–31]. In any case, systematic reviews and meta-analysis have shown that HIV self-testing can be successfully conducted by untrained users without in-person demonstrations [30]. Our observations emphasize the need to complete the classical paper instructions for use by other instructional tools such as short video film, which was preferred by 1 of 13 study participants for better instructions for use understanding. These findings are reminiscent to previous WHO recommendations for HIV self-test stating that all self-testers should have the possibility to access or receive assistance over the phone, through the internet, or with additional instructions such as video, animations, or diagrams [15].

## Substudy 2

All study participants carried out the COVID-19 self-test and succeeded in obtaining a valid test result with an overall usability index estimated at 98.5%. Some difficulty in the correct use of the pipette to transfer the blood sample was the principal reported concern encountered and was the most common reason for oral help. In previous reports on HIV self-testing, the difficulties in self-lancing and blood transfer to the cassette were also observed by lay users [32]. These features underline the importance of video instructions, when available. Although a small sample size of participants used the video instructions in this series, all of them not needed any help and used the pipette without any difficulty or error. The use of a hotline could also offer direct distant assistance.

## Substudy 3

The ability to correctly read and interpret the self-test results is considered as a critical step in self-testing [33]. This refers not only to the visual subjectivity related to good visual acuity (*i.e.* eye without illness) when reading and interpreting the results, but also to the number of bands to read on the test strip. Indeed, the Exacto® COVID-19 self-test (Biosynex Swiss SA) has three bands, one of which is for the internal control and two for the detection of IgG and IgM antibodies. The interpretation of a weak positive band may be therefore difficult for untrained users. In our series, the rate (98.8%) of correct interpretation of COVID-19 self-test results was high, as previously reported with HIV self-test using similar cassette [17, 18]. However, the majority (80%) of misinterpreted test results concerned a weak positive IgM band. This difficulty in reading some weak positive bands and in final interpretation of test results can even occur in lay users as well as trained-users during professional testing [34].

On the other hand, the interpretation of positive results with the serological IgM and IgG test of SARS-CoV-2 presents particularities in this period of the ongoing outbreak. While positive serology for other viral infections such as HIV means an active infection [35], a positive test result with the Exacto® COVID-19 self-test (Biosynex Swiss SA) rather indicates ongoing or previous SARS-CoV-2 infection, with serological immune IgG or IgM immune responses to SARS-CoV-2. Furthermore, according to the kinetic profile of the systemic humoral response against SARS-CoV-2 and the lifespan of circulating immunoglobulins, the presence of IgM alone or with IgG means that the contact with the virus was relatively recent [36]. The presence of IgG means that the contact with the virus occurred at least 14 days ago [36]. Thus, a positive test result on the COVID-19 self-test does not mean that the SARS-CoV-2 infection is still active. Despite the explanations were clearly given in the instructions for use, 10.2% of study participants were not aware that the COVID-19 self-test does not detect the presence of the virus. This misinterpretation of positive test results can provide unfortunate consequences

such as self-medication or psychological distress of variable intensity, especially in a person who has not received pre-test counseling [37].

## Substudy 4

The pre-test and post-test answers to the satisfaction questionnaire concerning the instructions for use (substudy 1), performing the self-test (substudy 2), and the interpretation of the results (substudy 2), showed that the large majority of the COVID-19 self-testing steps were considered easy by participants, as previously reported for HIV self-testing using similar rapid test cassette [17, 18]. However, the satisfaction with sample collection and blood transfer to the test cassette evolved from "very easy" in pre-test period to "rather easy" after having performed the self-test. This latter observation reminds us our previous experience with HIV self-testing, during which the fear of self-sticking provided capillary blood sample collection difficult in a minority of lay user [18].

## Strengths and limitations

Our study is original by highlighting for the first time the usability of COVID-19 self-test, as a novel approach to assess SARS-CoV-2-specific humoral immunity by using rapid diagnostic test and self-interpretation of the results. Our study also shows for the first time the possibility of correctly interpreting three bands on the strip of a rapid diagnostic test by lay users from general adult population. However, the study has some limitations. First, the presence of an observer may lead to a bias in our observations concerning the participants' ability to perform the tests and to interpret the results. Furthermore, the low sample size could reduce the study's power to detect a relative difference between groups with high precision. Finally, further steps are needed to improve mass screening for COVID-19, including the development of other tests such as oral fluid based self-testing, antigen self-testing, as well as home self-sampling.

The role of the COVID-19 self-test in fighting the epidemic, caring for infected people and preventing risk of transmission is not yet known. The possible risk of adverse effects of the COVID-19 self-test should not be underestimated, such as an individual assuming they are immune or non-contagious when they are not. This emphasizes the need for pre- and post-test counseling. Furthermore, there is limited understanding of adult public acceptability and usability of rapid diagnostic tests in the home setting, as most are currently designed as professional use to be carried out by healthcare professionals. Post-marketing surveillance for these potential adverse consequences will be needed. Nevertheless, the place of the COVID-19 self-test could simply be a complementary public health tool. Indeed, testing a large number of individuals for serological survey for example would be impractical if a blood sample is required for SARS-CoV-2 serologic testing in a laboratory. The solution to use self-sampling and self-testing with participants reporting their results to the clinicians or epidemiologists has been recently reported in a nationally representative serosurvey of SARS-CoV-2 in adults in England, demonstrating its feasibility [38].

According to the WHO [39], generalization of COVID-19 testing is key to controlling the spread of SARS-CoV-2 infection. In particular, the findings derived from serological assays can provide valuable information that would help to support the diagnosis, treatment and prevention of SARS-CoV-2 infection [40]. During the COVID-19 epidemic, novel approaches using individual involvement were proposed in addition to the collective public health approach, and both strategies were furthermore sometimes combined. For example, self-collected upper respiratory tract swabs for COVID-19 test has been shown as a feasible way to increase overall testing rate in South Africa [41], and the US Food and Drug Administration has approved the first kit for self-collected saliva specimen to be used for molecular testing of SARS-CoV-2 [42]. Self-

diagnosis of breathing complications from breathing sounds using the smartphone's microphone has been proposed as an appealing resolution for COVID-19 self-testing [43]. Self-reporting of an illness consistent with COVID-19 and artificial intelligence-coupled self-testing and tracking systems for COVID-19 have been developed using mobile phone applications [44, 45]. While the place of SARS-CoV-2-specific serology remains controversial [46, 47], the indications for the COVID-19 serological self-test have been the matter of poor attention from official agencies until now and remain to be defined [48]. It seems obvious that the motivations for carrying out a COVID-19 self-test would be clearly different than those which push to carry out an HIV self-test. The COVID-19 self-test allows an individual to test himself simply and quickly, without visiting a care structure, with the essential aim of knowing if the person is in the course of infection (presence of specific IgM alone) or had a past infection (presence of specific IgG, alone or associated with IgM). Thus, COVID-19 self-testing for serological screening could be proposed to identify exposed patients that are presumptively immune to SARS-CoV-2 secondary to ongoing or past-infection and to quantify the prevalence of exposure within a population for epidemiologic purposes. The instructions for use clearly explains that the lack of reactivity does not exclude the possibility of active SARS-CoV-2 infection and infectiousness in progress, and that in the presence of any IgG or IgM reactivities the patient must seek confirmatory antibody test by a clinical laboratory and clinical follow-up, which could contribute to a burden on the healthcare system, in particular during epidemic periods. It should be emphasized that it is not known if a positive antibody test represents protection and the concept of an "immunological passport" cannot be supported at this time [46, 47, 49]. Even if antibodies to SARS-CoV-2 are shown to offer some level of protection the durability of any such protection is not known at this time. Furthermore the presence of antibodies does not necessarily indicate that the person is not contagious particularly during the early phases of infection. It will therefore be important to pass this information on to subjects who self-test so that they continue to take precautions to protect themselves and others. While specific guidelines regarding how "presumptive immunity" will be determined and used do not exist, this potential use has probably generated the interest in the lay public [47]. In any case, an IgG positive COVID-19 self-test result may indicate recovery of a previous SARS-CoV-2 infection, even asymptomatic or mild. Interestingly, serological home testing could be associated with at-home saliva or swab self-sampling for further SARS-CoV-2 molecular diagnosis, and the widespread use of both home approaches is worthy of further study. Those whom the viral test indicates an active SARS-CoV-2 infection (including silent carriers and individuals with early or mild symptoms) will be able to take informed actions, such as self-isolation. Furthermore, the risk exposure of the healthy population may be mitigated by the actions taken by the (informed) infected population, thus slowing the spread of the coronavirus and flattening the curve. Importantly, a confirmed population of "recovered" individuals may facilitate many to return to work with no loss in protection for the most vulnerable. Recently, the British government, UK, has made SARS-CoV-2 home tests available to healthcare workers and the general public [50]. Home testing will be voluntary, but there is no doubt more people will test if the tests could be freely available.

Until a cure or a vaccine becomes available, antibody and viral testing for SARS-CoV-2 infection will play a critical role in limiting the pandemic and containing its economic damage. Our study demonstrates that COVID-19 self-testing for serological immune status assessment is highly feasible with potential for use by at least some groups with high levels of education. If deployed wisely, it may be complementary to other serological screening tools and could facilitate uptake of SARS-CoV-2 serology.

## Supporting information

**S1 Video. Video-based instruction for use of the Exacto® COVID-19 self-test (Biosynex Swiss SA).**
(MP4)

**S1 Appendix. Paper-based instruction for use of the Exacto® COVID-19 self-test (Biosynex Swiss SA).**
(TIF)

**S2 Appendix. Study questionnaires in French (original language).**
(DOCX)

**S3 Appendix. Study questionnaires in English.**
(DOCX)

## Acknowledgments

The authors are grateful to the volunteers for their willingness to participate in the study. We also thank Biosynex, Strasbourg, France, for providing the Exacto® COVID-19 self-tests for the study.

## Author Contributions

**Conceptualization:** Serge Tonen-Wolyec, Raphael Dupont, Laurent Bélec.

**Data curation:** Raphael Dupont.

**Formal analysis:** Serge Tonen-Wolyec.

**Investigation:** Raphael Dupont, Laurent Bélec.

**Methodology:** Serge Tonen-Wolyec, Salomon Batina-Agasa, Marie-Pierre Hayette, Laurent Bélec.

**Supervision:** Serge Tonen-Wolyec, Raphael Dupont, Salomon Batina-Agasa, Marie-Pierre Hayette, Laurent Bélec.

**Validation:** Serge Tonen-Wolyec, Laurent Bélec.

**Visualization:** Serge Tonen-Wolyec, Laurent Bélec.

**Writing – original draft:** Serge Tonen-Wolyec, Laurent Bélec.

**Writing – review & editing:** Serge Tonen-Wolyec, Raphael Dupont, Salomon Batina-Agasa, Marie-Pierre Hayette, Laurent Bélec.

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
