## [Decision Letter · Decision Letter 0]

21 Aug 2020

PONE-D-20-20619

Capillary whole-blood IgG-IgM COVID-19 self-test  as a serological screening tool for SARS-CoV-2 infection  adapted to the general public

PLOS ONE

Dear Dr. Belec

Thank you for submitting your manuscript to PLOS ONE. After careful consideration, we feel that it has merit but does not fully meet PLOS ONE’s publication criteria as it currently stands. Therefore, we invite you to submit a revised version of the manuscript that addresses the points raised during the review process.

Please address the critique of both reviewers

We look forward to receiving your revised manuscript.

Kind regards,

Alan Landay

Academic Editor

PLOS ONE

Journal Requirements:

2. Please include additional information regarding the survey or questionnaire used in the study and ensure that you have provided sufficient details that others could replicate the analyses.

For instance, if you developed a questionnaire as part of this study and it is not under a copyright more restrictive than CC-BY, please include a copy, in both the original language and English, as Supporting Information.

'Dr. Serge Tonen-Wolyec was recipient of ERASMUS+ program between the University of Kisangani, Democratic Republic of the Congo, and the University of Liège, Belgium.'

'The authors received no specific funding for this work.'

'The authors have declared that no competing interests exist.'

We note that one or more of the authors are employed by a commercial company: BioSynex

Reviewers' comments:

Reviewer's Responses to Questions

**Comments to the Author**

1. Is the manuscript technically sound, and do the data support the conclusions?

Reviewer #1: Partly

Reviewer #2: Partly

2. Has the statistical analysis been performed appropriately and rigorously? 

Reviewer #1: Yes

Reviewer #2: Yes

3. Have the authors made all data underlying the findings in their manuscript fully available?

Reviewer #1: Yes

Reviewer #2: Yes

4. Is the manuscript presented in an intelligible fashion and written in standard English?

Reviewer #1: No

Reviewer #2: Yes

5. Review Comments to the Author

Reviewer #1: This study is potentially one of several necessary but not sufficient steps towards translation to practice. However, the discussion must be made much more conservative. The extensive speculation on the role of home serology testing could create safety problems and is of major concern.

Highlight [page 8]: 98.5% (95% CI: 96.5–99.4) test results were correctly interpreted, while misinterpretation occurred in only…

Note [page 8]: L47. What is the definition of the correct interpretation of the test?

Note [page 10]: L88 Change ‘as’ to ‘as well as’

Highlight [page 10]: HIV self-testing has demonstrated high acceptability with very convenient usability in various adolescent and adult profane populations from developed as resources- constrained settings [17-21].

Note [page 10]: L88 profane? Don’t think you mean this- suggest remove this word.

Highlight [page 11]: The BIOSYNEX ®COVID-19 BSS (IgG/IgM) (Biosynex Swiss SA) showed sensitivity of 97.4% and specificity of 100%, demonstrating high analytical performances allowing convenient management of suspected on-going and past-infections.

Note [page 11]: L 119: Have these results been peer reviewed and published elsewhere? If so please provide reference? Why not publish the this study and the performance characteristics of the test in the same paper? They ideally need to be assessed together.

Highlight [page 11]: The online instruction in the video for use was available online from Youtube [24].

Note [page 11]: When the QR code on Figure 1 is scanned it says the video has been taken down. Please provide the video or QR code. Ideally the video could be permanently attached to this paper by the journal rather than relying on a Youtube video that could be taken down again.

Note [page 12]: 132 See latter suggestions about moving full instructions to supplementary materials and using just top half of interpretation panel as Fig 1. Legend needs to state that this was the exact instructions provided to the subjects in this study in both legends.

Note [page 12]: L 134: simplify this phrase

Highlight [page 12]: of the Exacto® COVID-19 self-test (Biosynex Swiss SA) is a cross-sectional study performed between April and May 2020 by home-based recruitment of adult volunteers using a door-to- door community approach, in 15 neighborhoods of Strasbourg and its suburbs,…

Note [page 12]: How were these neighborhoods selected? Was there a wide range of socio-economic and eductaional status and was this representative of developed countries in Northern Europe? Will need a discussion on how generalizable are these results likely to be.

Note [page 14]: L189: Change appeal for to provide

Highlight [page 14]: The observer was responsible for recording the respect or not of each step, appeal for verbal assistance (mimicking telephone support), difficulty, and errors on a standardized sheet.

Note [page 14]: L196: change proposed to provided

Highlight [page 14]: In a private setting supervised by an observer, eight standardized test results including four positive tests (one weak positive for IgM, one clearly positive for IgM, one clearly positive for IgG but weak positive for IgM, and one clearly positive for IgM and IgG), two negative tests, and two invalid tests were proposed to the participants for interpretation after successive…

Note [page 14]: L196: delete successive

Note [page 14]: L201- 202: suggest change No to #

Highlight [page 14]: Panel of 8 Exacto® COVID-19 self-test (Biosynex Swiss SA) cassettes, including 4 positive tests (n°2, n°3, n°6 and n°7), 2 negative tests (n°2 and n°7) and 2 invalid tests (n°1 and n°5).

Note [page 14]: L202: change successively to randomly

Highlight [page 14]: Each volunteer successively drew 4 tests among a panel of 8 and interpreted them with the help of the reading and interpretation scale.

Note [page 15]: L228: change and to or

Highlight [page 15]: excluded because they were trained (n=12), less than 18 years old (n=5), and not consenting (n=10).

Note [page 18]: L277: delete HIV

Highlight [page 18]: Overall, the mean time of HIV self-test performance (since the opening of the box until the migration step) was…

Note [page 19]: L293: delete successive- not clear what this means

Highlight [page 19]: COVID-19 self-test results after successive random selection of four tests from a panel of eight standardized tests.

Note [page 20]: L308: would be interesting to know if there were any differences in the results from substudy 4 for those previously in subsidy 2 versus 3.

Highlight [page 20]: Substudy 4.

Note [page 22]: L349 Europeans of high educational attainment

Highlight [page 22]: Finally, our observations lay the foundations for the potential large-scale use of COVID-19 self-test in lay adults, at least Europeans, to complete the arsenal of available serological tests used to assess the immune status vis-a-vis SARS-CoV-2.

Note [page 23]: L 376: ..error, however numbers in this group were small.

Highlight [page 23]: In the present series, all participants using the video instructions did not need any help and used the pipette without any difficulty or error.

Note [page 24]: L379: Change delicate to critical

Highlight [page 24]: considered as a delicate step in self-testing [34].

Note [page 24]: L395 to 396: Is this really established for SARS-CoV-2 infection. Please provide references

Highlight [page 24]: Furthermore, the presence of IgM alone or with IgG means that the contact with the virus was relatively recent.

Note [page 24]: L401: Change neuropsychiatric disorders to psychological distress and not psychologically prepared to who has not received pre-test counseling.

Highlight [page 24]: This misinterpretation of positive test results can provide unfortunate consequences such as self-medication or neuro-psychiatric disorders of variable intensity, especially in a person not psychologically prepared [38].

Note [page 25]: L419: limit the study’s power to detect….what?

Highlight [page 25]: Furthermore, the low sample size could reduce the study’s power.

Note [page 25]: L425: novel rather than original

Highlight [page 25]: During the COVID-19 epidemic, original approaches using individual involvement were proposed in addition to the collective public health approach, and both strategies were furthermore sometimes combined.

Note [page 26]: L439: suggest delete ‘, but this….study”

Highlight [page 26]: It seems obvious that the motivations for carrying out a COVID-19 self-test would be clearly different than those which push to carry out an HIV self-test, but this problematic exceeds the aim of our study.

Note [page 26]: L442: change has made too had

Highlight [page 26]: The COVID-19 self-test allows an individual to test himself simply and quickly, without visiting a care structure, with the essential aim of knowing if the person is in the course of infection (presence of specific IgM alone) or has made a past infection (…

Note [page 26]: L443: Need to emphasize that it is not yet known if antibodies are protective and if so how durable this protection is and if antibodies guarantee they cannot infect others. Must emphasize the importance of conveying this to the subjects self-testing and of their need to continue to take precautions to protect themselves and others.

Highlight [page 26]: Thus, COVID-19 self-testing for serological screening could be proposed to identify exposed patients that are presumptively immune to SARS -CoV- 2 secondary to ongoing or past-infection and to quantify the prevalence of exposure within a population for epi…

Note [page 26]: L448: “refer..” change to seek confirmatory antibody test by a clinical laboratory and clinical follow-up” Need to comment on the burden this will place on the health care system.

Highlight [page 26]: The instructions for use clearly explains that the lack of reactivity does not eliminate a SARS-CoV-2 infection in progress, and that in the presence of any IgG or IgM reactivities the patient must refer to a health care structure for clinical…

Note [page 26]: L449: change to It should be emphasized that it is not known if a positive antibody test represents protection and the concept of an “immunological passport” cannot be supported at this time.

Highlight [page 26]: In any case, the presence of reactivities could constitute an "immunological passport" of protection [46,47], because it is not known if anti-SARS-CoV-2 antibodies are protective at this time, although the general assumption is that the presence of antibod…

Note [page 27]: L454 Change most excitement to interest

Highlight [page 27]: “presumptive immunity” will be determined and used do not exist, this potential use has probably generated the most excitement in the lay public [47].

Note [page 27]: L456-457: delete: …and would…individuals” No evidence to support his statement.

Highlight [page 27]: In any case, an IgG positive COVID-19 self-test result may indicate recovery of a previous SARS-CoV-2 infection, even asymptomatic or mild, and would allow to take more moderate precautions and also to comfortably interact with other COVID-19-seropositive individuals.

Note [page 27]: L459: delete would be hugely beneficial to public health. The is conjecture. Suggest ‘is worthy of further study’

Highlight [page 27]: Interestingly, serological home testing could be associated with at-home saliva or swab self-sampling for further SARS- CoV-2 molecular diagnosis, and the widespread use of both home approaches would be hugely beneficial to public health.

Note [page 27]: L461: should consider themselves potentially infected and self-isolate until the results of clinical testing for the virus is known.

Highlight [page 27]: Those whom the viral test indicates an active SARS-CoV-2 infection (including silent carriers and patients with early or mild symptoms) will be able to take informed actions, such as self-isolation.

Note [page 27]: L465: Change ‘would allow’ to ‘may facilitate’. All of this discussion is too much conjecture and should be toned down.

Highlight [page 27]: Importantly, a confirmed population of “recovered” individuals would allow many to return to work, lead to partial lifting of “stay

Note [page 27]: L451: change will to may and indicate how this could be study to support such policies. Discuss how cost-effectiveness would have to be studies.

Highlight [page 27]: Removing financial barriers to self-testing by making publicly-funded tests available free to the entire population will help maximize rapid implementation and help COVID-19-affected country to recover and get back to work.

Note [page 27]: L476: change the general public to ‘by at least some groups with high levels of education.

Highlight [page 27]: Our features demonstrate that COVID-19 self-testing for serological immune status assessment is highly feasible with potential for use by the general public.

Note [page 27]: L477: change will to may

Highlight [page 27]: If deployed wisely, it will be complementary to other serological screening tools and

Note [page 28]: L478: change ‘offer an immediate and easy solution for’ to facilitate uptake of SARS-CoV-2 serology and delete rest of sentence.

Highlight [page 28]: could offer an immediate and easy solution for SARS-CoV-2 serology, especially during recovery or de-confinement periods.

Note [page 33]: Figure 1. Impractical to include the entire instruction in the main body of the paper. It should be moved to supplementary materials. The top half of the interpretation panel with an appropriate legend would be more appropriate. Given this is the peer reviewed study examining the issue of interpretation the comment under performance about the 98.5% correct interpretation should be removed. Also the reference to support the performance characteristics of the test shown above that statement needs to be provided.

Highlight [page 34]: Click here to access/download;Figure;Fig…

None of these links worked on this version.

Reviewer #2: The authors report on the practicability of a prototype capillary whole-blood IgG-IgM COVID-19 self-test (Exacto COVID-19 self test, Biosynex Swiss, SA,Freiburg, Switzerland) as a serological screening tool for SARS-COV-2 infection adapted to the general public. They performed their evaluation of this test using a cross sectional, general adult population study between April and May 2020 in Strasbourg, France. The study design consisted of face-to-face, paper-based, semi-structured, and self administrated questionnaires. The study enrolled 167 participants of which 82% had a post-graduate level of education. The study evaluated the participants ability to use the test in a number of different testing settings. The authors conclude that 100% of the participants found that performing the self test was easy and 98% found that the interpretation of the self-test results are easy.

While this study is very interesting and brings forward an important POC / self- administered SARS-COv-2 serological assay the authors failed to bench mark the antibody status to a gold standard lab based assay. The absence of this weakens their initial pilot findings. Does it bring value if people can follow directions and get a result if the test does not corelate highly to what would be considered a typical bench mark to an assay performed in the laboratory under a clinical standard? The absence of comparative data is a major flaw in the study design.

6. PLOS authors have the option to publish the peer review history of their article (what does this mean?). If published, this will include your full peer review and any attached files.

Reviewer #1: No

Reviewer #2: No

---

## [Author Response · Author response to Decision Letter 0]

11 Sep 2020

Responses to journal requirements and to Reviewers 

Journal Requirements:

https://clicktime.symantec.com/3Ab2UDzwphFJFJ5wTH8Dthe6H2?u=https%3A%2F%2Fjournals.plos.org%2Fplosone%2Fs%2Ffile%3Fid%3DwjVg%2FPLOSOne_formatting_sample_main_body.pdf and

https://clicktime.symantec.com/3J1bpueumkNeCCUwpeXGyX66H2?u=https%3A%2F%2Fjournals.plos.org%2Fplosone%2Fs%2Ffile%3Fid%3Dba62%2FPLOSOne_formatting_sample_title_authors_affiliations.pdf

Our answer: We have checked that the manuscript meets the PLOS ONE’S requirements, including file names and affiliations. 

2. Please include additional information regarding the survey or questionnaire used in the study and ensure that you have provided sufficient details that others could replicate the analyses.

For instance, if you developed a questionnaire as part of this study and it is not under a copyright more restrictive than CC-BY, please include a copy, in both the original language and English, as Supporting Information.

Our answer: As requested, the study questionnaires in French (original language) as well as in English have been uploaded in the submission system, as supporting information.

'Dr. Serge Tonen-Wolyec was recipient of ERASMUS+ program between the University of Kisangani, Democratic Republic of the Congo, and the University of Liège, Belgium.'

'The authors received no specific funding for this work.'

Our answer: In order to acknowledge the journal requirement, we have removed any funding-related test from the manuscript and we have updated our Funding Statement as follow: “This work was partly supported by Biosynex SA. The funders played a role in providing the prototype SARS-CoV-2 test for self-test (Exacto® COVID-19 self-test, Biosynex Swiss SA) and data collection. The study design, analysis, decision to publish, and preparation of the manuscript were not sponsored. Biosynex SA also provided support for this study in the form of salary for Dr. Raphael Dupont. The specific role of this author is articulated in the ‘author contributions’ section. Dr. Serge Tonen-Wolyec was recipient of ERASMUS+ program between the University of Kisangani, Democratic Republic of the Congo, and the University of Liège, Belgium. There was no additional external funding received for this study.” 

Our answer: We have included our amended Funding statement within our cover letter. 

'The authors have declared that no competing interests exist.'

We note that one or more of the authors are employed by a commercial company: BioSynex

Our answer: The authors have read the journal’s policy and have the following competing interests: Dr. Raphael Dupont is a paid employee of Biosynex SA. The authors would like to declare the following patents/patent applications associated with this research: https://bases-marques.inpi.fr/Typo3_INPI_Marques/ajoutListe?page=1&idObjet=1484785_202032_tmint&scroll=462.4761962890625. This does not alter our adherence to PLOS ONE policies on sharing data and materials. We have added this highlighting in our cover letter and online submission. 

5. PLOS requires an ORCID iD for the corresponding author in Editorial Manager on papers submitted after December 6th, 2016. Please ensure that you have an ORCID iD and that it is validated in Editorial Manager. To do this, go to ‘Update my Information’ (in the upper left-hand corner of the main menu), and click on the Fetch/Validate link next to the ORCID field. This will take you to the ORCID site and allow you to create a new iD or authenticate a pre-existing iD in Editorial Manager. Please see the following video for instructions on linking an ORCID iD to your Editorial Manager account: https://clicktime.symantec.com/3QJAoi3RwJ4rwEt9UViL8wS6H2?u=https%3A%2F%2Fwww.youtube.com%2Fwatch%3Fv%3D_xcclfuvtxQ

Our answer: We have added a validated ORCID iD (https://orcid.org/0000-0002-5001-0405) of the corresponding author in Editorial Manager. 

Reviewers' comments: 

Reviewer's Responses to Questions

Comments to the Author

1. Is the manuscript technically sound, and do the data support the conclusions?

Reviewer #1: Partly

Reviewer #2: Partly

Our answer: We thank the reviewers for their nice comments on our work. However, in order to acknowledge the comments raised by referees, we have made corrections thorough the manuscript; therefore, we hope that our revised manuscript is more technically sound.

2. Has the statistical analysis been performed appropriately and rigorously?

Reviewer #1: Yes

Reviewer #2: Yes

Our answer: We thank the reviewers for their nice comments on our work. 

3. Have the authors made all data underlying the findings in their manuscript fully available?

Reviewer #1: Yes

Reviewer #2: Yes

Our answer: We thank the reviewers for their nice comments on our work. 

4. Is the manuscript presented in an intelligible fashion and written in standard English?

Reviewer #1: No

Reviewer #2: Yes

Our answer: In order to acknowledge the comments raised by Reviewer # 1, we have corrected words and grammar as suggested by Referee. We hope that our revised manuscript is presented in an intelligible fashion and written in standard American English. 

5. Review Comments to the Author

Answer to reviewer #1

This study is potentially one of several necessary but not sufficient steps towards translation to practice. However, the discussion must be made much more conservative. The extensive speculation on the role of home serology testing could create safety problems and is of major concern.

Our answer: The remark of the reviewer is right. To acknowledge the reviewer’s concern, we have completed the Strengths and limitations section by adding the following paragraph: “The role of the COVID-19 self-test in fighting the epidemic, caring for infected people and preventing risk of transmission is not yet known. The possible risk of adverse effects of the COVID-19 self-test should not be underestimated, such as a pseudo-insurance of immunity or non-contagiousness. Furthermore, there is limited understanding of adult public acceptability and usability of rapid diagnostic tests in the home setting, as most are currently designed as professional use to be carried out by healthcare professionals. It will of course be necessary to precisely assess all these potential perverse effects. However, the place of the COVID-19 self-test could simply be a complementary public health tool. Indeed, testing a large number of individuals for serological survey for example would be impractical if a blood sample is required for SARS-CoV-2 serologic testing in a laboratory. The solution to use self-sampling and self-testing with participants reporting their results to the clinicians or epidemiologists has been recently reported in a nationally representative serosurvey of SARS-CoV-2 in adults in England, demonstrating its full feasibility [Atchison et al., 2020].”

Atchison C, Pristerà P, Cooper E, Papageorgiou V, Redd R, Piggin M, Flower B, Fontana G, Satkunarajah S, Ashrafian H, Lawrence-Jones A, Naar L, Chigwende J, Gibbard S, Riley S, Darzi A, Elliott P, Ashby D, Barclay W, Cooke GS, Ward H. Usability and acceptability of home-based self-testing for SARS-CoV-2 antibodies for population surveillance. Clin Infect Dis. 2020 Aug 12:ciaa1178. doi: 10.1093/cid/ciaa1178.

Highlight [page 8]: 98.5% (95% CI: 96.5–99.4) test results were correctly interpreted, while misinterpretation occurred in only…

Note [page 8]: L47. What is the definition of the correct interpretation of the test?

Our answer: Since the expected results were known from the code numbers of the eight standardized tests, the correct interpretation of the tests was defined by the percent agreement between the tests results read and interpreted by the participants compared to the expected results coded by the numbers and verified by observers. Thus, misinterpretation corresponded to the percent disagreement between the test results read and interpret by the participants and the expected results coded by the numbers. We have added these clarifications in the abstract and the body of the text. 

Note [page 10]: L88 Change ‘as’ to ‘as well as’

Highlight [page 10]: HIV self-testing has demonstrated high acceptability with very convenient usability in various adolescent and adult profane populations from developed as resources- constrained settings [17-21].

Note [page 10]: L88 profane? Don’t think you mean this- suggest remove this word.

Our answer: We have corrected the sentence, as suggested.

Highlight [page 11]: The BIOSYNEX ®COVID-19 BSS (IgG/IgM) (Biosynex Swiss SA) showed sensitivity of 97.4% and specificity of 100%, demonstrating high analytical performances allowing convenient management of suspected on-going and past-infections.

Note [page 11]: L 119: Have these results been peer reviewed and published elsewhere? If so please provide reference? Why not publish the this study and the performance characteristics of the test in the same paper? They ideally need to be assessed together.

Our answer: While the purpose of our study was not to assess the virological analytical performances of the BIOSYNEX® COVID-19 BSS (IgG/IgM) (Biosynex Swiss SA), this rapid diagnostic test has been fully recommended for both SARS-CoV-2-specific IgG and IgM detection by the French Ministry of Health (https://covid-19.sante.gouv.fr/tests. ; last access 25 August 2020), following an official report from the National Reference Center for Respiratory Viruses [Centre National de Référence Virus des infection respiratoires (dont la grippe)], Institut Pasteur, Paris. We have added this information in the text. 

Highlight [page 11]: The online instruction in the video for use was available online from Youtube [24].

Note [page 11]: When the QR code on Figure 1 is scanned it says the video has been taken down. Please provide the video or QR code. Ideally the video could be permanently attached to this paper by the journal rather than relying on a Youtube video that could be taken down again.

Note [page 12]: 132 See latter suggestions about moving full instructions to supplementary materials and using just top half of interpretation panel as Fig 1. Legend needs to state that this was the exact instructions provided to the subjects in this study in both legends.

Our answer: In order to acknowledge the comments raised by Reviewer # 1, we have moved the full instruction for use to supplementary materials. Furthermore, we have provided the video instruction as its supporting information file. 

Note [page 12]: L 134: simplify this phrase

Highlight [page 12]: of the Exacto® COVID-19 self-test (Biosynex Swiss SA) is a cross-sectional study performed between April and May 2020 by home-based recruitment of adult volunteers using a door-to- door community approach, in 15 neighborhoods of Strasbourg and its suburbs,…

Our answer: We have simplified this sentence as suggested. 

Note [page 12]: How were these neighborhoods selected? Was there a wide range of socio-economic and eductaional status and was this representative of developed countries in Northern Europe? Will need a discussion on how generalizable are these results likely to be.

Our answer: Due to the limited movement during the lockdown period, the choice of these neighborhoods and its suburbs was based on their easy accessibility and their high prevalence of reported cases of SARS-CoV-2 infection. We have added this sentence in the “Study design and recruitment of participants” section for more highlighting. 

Note [page 14]: L189: Change appeal for to provide

Highlight [page 14]: The observer was responsible for recording the respect or not of each step, appeal for verbal assistance (mimicking telephone support), difficulty, and errors on a standardized sheet.

Our answer: We have changed the words as suggested.

Note [page 14]: L196: change proposed to provided

Highlight [page 14]: In a private setting supervised by an observer, eight standardized test results including four positive tests (one weak positive for IgM, one clearly positive for IgM, one clearly positive for IgG but weak positive for IgM, and one clearly positive for IgM and IgG), two negative tests, and two invalid tests were proposed to the participants for interpretation after successive…

Our answer: We have changed the words as suggested. 

Note [page 14]: L196: delete successive

Note [page 14]: L201- 202: suggest change No to #

Highlight [page 14]: Panel of 8 Exacto® COVID-19 self-test (Biosynex Swiss SA) cassettes, including 4 positive tests (n°2, n°3, n°6 and n°7), 2 negative tests (n°2 and n°7) and 2 invalid tests (n°1 and n°5).

Our answer: We have changed the words as suggested. 

Note [page 14]: L202: change successively to randomly

Highlight [page 14]: Each volunteer successively drew 4 tests among a panel of 8 and interpreted them with the help of the reading and interpretation scale.

Our answer: We have changed the words as suggested.

Note [page 15]: L228: change and to or

Highlight [page 15]: excluded because they were trained (n=12), less than 18 years old (n=5), and not consenting (n=10).

Our answer: We have changed the words as suggested. 

Note [page 18]: L277: delete HIV

Highlight [page 18]: Overall, the mean time of HIV self-test performance (since the opening of the box until the migration step) was…

Our answer: It was a mistake, we have deleted it. 

Note [page 19]: L293: delete successive- not clear what this means

Highlight [page 19]: COVID-19 self-test results after successive random selection of four tests from a panel of eight standardized tests.

Our answer: We have deleted it. 

Note [page 20]: L308: would be interesting to know if there were any differences in the results from substudy 4 for those previously in subsidy 2 versus 3.

Highlight [page 20]: Substudy 4.

Our answer: We did not assess such comparisons.

Note [page 22]: L349 Europeans of high educational attainment

Highlight [page 22]: Finally, our observations lay the foundations for the potential large-scale use of COVID-19 self-test in lay adults, at least Europeans, to complete the arsenal of available serological tests used to assess the immune status vis-a-vis SARS-CoV-2.

Our answer: We thank the reviewer for this clarification, which we have added to the text.

Note [page 23]: L 376: ..error, however numbers in this group were small.

Highlight [page 23]: In the present series, all participants using the video instructions did not need any help and used the pipette without any difficulty or error.

Our answer: We have added this precision related to the small sample size in this discussion as follows: “Although a small sample size of participants used the video instructions in this series, all of them did not need any help and used the pipette without any difficulty or error.” 

Note [page 24]: L379: Change delicate to critical

Highlight [page 24]: considered as a delicate step in self-testing [34].

Our answer: We have changed the words as suggested.

Note [page 24]: L395 to 396: Is this really established for SARS-CoV-2 infection. Please provide references

Highlight [page 24]: Furthermore, the presence of IgM alone or with IgG means that the contact with the virus was relatively recent.

Our answer: To acknowledge the reviewer’s remarks, we completed as follows: “Furthermore, according to the kinetic profile of the systemic humoral response against SARS-CoV-2 and the lifespan of circulating immunoglobulins, the presence of IgM alone or with IgG means that the contact with the virus was relatively recent [37]”.

Note [page 24]: L401: Change neuropsychiatric disorders to psychological distress and not psychologically prepared to who has not received pre-test counseling.

Highlight [page 24]: This misinterpretation of positive test results can provide unfortunate consequences such as self-medication or neuro-psychiatric disorders of variable intensity, especially in a person not psychologically prepared [38].

Our answer: We have changed the sentence as suggested. 

Note [page 25]: L419: limit the study’s power to detect….what?

Highlight [page 25]: Furthermore, the low sample size could reduce the study’s power.

Our answer: The low sample size could reduce the study’s power to detect a relative difference between groups with high precision. 

Note [page 25]: L425: novel rather than original

Highlight [page 25]: During the COVID-19 epidemic, original approaches using individual involvement were proposed in addition to the collective public health approach, and both strategies were furthermore sometimes combined.

Our answer: We have changed the words as suggested. 

Note [page 26]: L439: suggest delete ‘, but this….study”

Highlight [page 26]: It seems obvious that the motivations for carrying out a COVID-19 self-test would be clearly different than those which push to carry out an HIV self-test, but this problematic exceeds the aim of our study.

Our answer: We have deleted the sentence as suggested.

Note [page 26]: L442: change has made too had

Highlight [page 26]: The COVID-19 self-test allows an individual to test himself simply and quickly, without visiting a care structure, with the essential aim of knowing if the person is in the course of infection (presence of specific IgM alone) or has made a past infection (…

Our answer: We have changed the words as suggested. 

Note [page 26]: L443: Need to emphasize that it is not yet known if antibodies are protective and if so how durable this protection is and if antibodies guarantee they cannot infect others. Must emphasize the importance of conveying this to the subjects self-testing and of their need to continue to take precautions to protect themselves and others.

Highlight [page 26]: Thus, COVID-19 self-testing for serological screening could be proposed to identify exposed patients that are presumptively immune to SARS -CoV- 2 secondary to ongoing or past-infection and to quantify the prevalence of exposure within a population for epi…

Our answer: To acknowledge these reviewer’s remarks, we have added the following sentence: “However, it should be emphasized that the level of protection of seropositivity for SARS-CoV-2 as well as its duration are not known, and even that the presence of specific antibodies does not mean that the person is not contagious, particularly in onset of infection. It will therefore be important to pass this information on to subjects who self-test so that they continue to take precautions to protect themselves and others.”

Note [page 26]: L448: “refer..” change to seek confirmatory antibody test by a clinical laboratory and clinical follow-up” Need to comment on the burden this will place on the health care system. 

Highlight [page 26]: The instructions for use clearly explains that the lack of reactivity does not eliminate a SARS-CoV-2 infection in progress, and that in the presence of any IgG or IgM reactivities the patient must refer to a health care structure for clinical…

Our answer: We have changed the sentence as suggested, and added that “which could contribute to accentuating tensions in the healthcare system, in particular during epidemic periods”. 

Note [page 26]: L449: change to It should be emphasized that it is not known if a positive antibody test represents protection and the concept of an “immunological passport” cannot be supported at this time.

Highlight [page 26]: In any case, the presence of reactivities could constitute an "immunological passport" of protection [46,47], because it is not known if anti-SARS-CoV-2 antibodies are protective at this time, although the general assumption is that the presence of antibod…

Our answer: We have deleted the ambiguous sentence: “…..because it is not known if anti-SARS-CoV-2 antibodies are protective at this time…..”.

Note [page 27]: L454 Change most excitement to interest

Highlight [page 27]: “presumptive immunity” will be determined and used do not exist, this potential use has probably generated the most excitement in the lay public [47].

Our answer: We have changed the words as suggested. 

Note [page 27]: L456-457: delete: …and would…individuals” No evidence to support his statement.

Highlight [page 27]: In any case, an IgG positive COVID-19 self-test result may indicate recovery of a previous SARS-CoV-2 infection, even asymptomatic or mild, and would allow to take more moderate precautions and also to comfortably interact with other COVID-19-seropositive individuals.

Our answer: We have deleted it as suggested. 

Note [page 27]: L459: delete would be hugely beneficial to public health. The is conjecture. Suggest ‘is worthy of further study’

Highlight [page 27]: Interestingly, serological home testing could be associated with at-home saliva or swab self-sampling for further SARS- CoV-2 molecular diagnosis, and the widespread use of both home approaches would be hugely beneficial to public health.

Our answer: We have corrected the sentence as suggested.

Note [page 27]: L461: should consider themselves potentially infected and self-isolate until the results of clinical testing for the virus is known.

Highlight [page 27]: Those whom the viral test indicates an active SARS-CoV-2 infection (including silent carriers and patients with early or mild symptoms) will be able to take informed actions, such as self-isolation.

Our answer: We agree with the reviewer. We have changed “patients” by “individuals”.

Note [page 27]: L465: Change ‘would allow’ to ‘may facilitate’. All of this discussion is too much conjecture and should be toned down.

Highlight [page 27]: Importantly, a confirmed population of “recovered” individuals would allow many to return to work, lead to partial lifting of “stay

Our answer: We have corrected the sentence as suggested. 

Note [page 27]: L451: change will to may and indicate how this could be study to support such policies. Discuss how cost-effectiveness would have to be studies.

Highlight [page 27]: Removing financial barriers to self-testing by making publicly-funded tests available free to the entire population will help maximize rapid implementation and help COVID-19-affected country to recover and get back to work.

Our answer: We have deleted this ambiguous sentence.

Note [page 27]: L476: change the general public to ‘by at least some groups with high levels of education.

Highlight [page 27]: Our features demonstrate that COVID-19 self-testing for serological immune status assessment is highly feasible with potential for use by the general public.

Our answer: We have changed the sentence as suggested. 

Note [page 27]: L477: change will to may

Highlight [page 27]: If deployed wisely, it will be complementary to other serological screening tools and

Our answer: We have changed the word as suggested.

Note [page 28]: L478: change ‘offer an immediate and easy solution for’ to facilitate uptake of SARS-CoV-2 serology and delete rest of sentence.

Highlight [page 28]: could offer an immediate and easy solution for SARS-CoV-2 serology, especially during recovery or de-confinement periods.

Our answer: We have changed the sentence as suggested. 

Note [page 33]: Figure 1. Impractical to include the entire instruction in the main body of the paper. It should be moved to supplementary materials. The top half of the interpretation panel with an appropriate legend would be more appropriate. Given this is the peer reviewed study examining the issue of interpretation the comment under performance about the 98.5% correct interpretation should be removed. Also the reference to support the performance characteristics of the test shown above that statement needs to be provided.

Highlight [page 34]: Click here to access/download;Figure;Fig…

None of these links worked on this version.

Our answer: As answered above, we have moved the full instruction for use to supplementary materials. And we have provided the video instruction as its supporting information file. However, we have added a Section A to the former Figure 3 (considered as a Section B) to present the interpretation of the results. Thus, this new figure is entitled Fig 2 in the revised version of our manuscript with legend written as follows: “Fig 2. Interpretation of self-test results. A. The self-test result was interpreted as negative when a Control line (C) was present and readable and the “IgG” and “IgM” lines were absent. It was positive when a “C” and “IgM” (clearly or poorly readable) (case 1), or “C” and “IgG”, or “C”, “IgM” (clearly or poorly readable), and “IgG” lines were present. Finally, it was invalid when the “C” line was absent regardless of the presence or absence of the “IgG” and/or “IgM” line. B. Panel of 8 Exacto® COVID-19 self-test (Biosynex Swiss SA) cassettes, including 4 positive tests (#2, #3, #6 and #7), 2 negative tests (#4 and #8) and 2 invalid tests (#1 and #5). The #2 and #37 are weakly positive for IgM. Each volunteer randomly drew 4 tests among a panel of 8 and interpreted them with the help of the reading and interpretation scale. The observer noted the number of the drawn test and the result given by the participant”. 

Concerning the interpretation of results, since the expected results were known from the code numbers of the eight standardized tests, the correct interpretation of the tests was defined by the percent agreement between the tests results read and interpret by the participants compared to the expected results coded by the numbers and verified by observers. Thus, misinterpretation corresponded to the percent disagreement between the test results read and interpret by the participants and the expected results coded by the numbers. We have added these clarifications in the abstract and the body of the text.

Finally, the virological analytical performances characteristic of the evaluated self-test are provided in the Material and methods section, in the Prototype SARS-CoV-2 test for self-testing. 

Answer to reviewer #2

The authors report on the practicability of a prototype capillary whole-blood IgG-IgM COVID-19 self-test (Exacto COVID-19 self test, Biosynex Swiss, SA,Freiburg, Switzerland) as a serological screening tool for SARS-COV-2 infection adapted to the general public. They performed their evaluation of this test using a cross sectional, general adult population study between April and May 2020 in Strasbourg, France. The study design consisted of face-to-face, paper-based, semi-structured, and self administrated questionnaires. The study enrolled 167 participants of which 82% had a post-graduate level of education. The study evaluated the participants ability to use the test in a number of different testing settings. The authors conclude that 100% of the participants found that performing the self test was easy and 98% found that the interpretation of the self-test results are easy.

Our answer: We thank the reviewer for this perfect summary of our study. 

While this study is very interesting and brings forward an important POC / self- administered SARS-COv-2 serological assay the authors failed to bench mark the antibody status to a gold standard lab based assay. The absence of this weakens their initial pilot findings. Does it bring value if people can follow directions and get a result if the test does not corelate highly to what would be considered a typical bench mark to an assay performed in the laboratory under a clinical standard? The absence of comparative data is a major flaw in the study design.

Our answer: We thank the reviewer for this pertinent remark. However, the objective of this survey was to assess the ability of lay persons to perform or interpret a serological test for SARS-CoV-2 immunochromatography. It was not intended to conduct a self-test performance study as such a study would require a large enough sample size of positive individuals to properly estimate the sensitivity of the self-test. Although this survey was carried out during the epidemic period in France, it should be noted that at that time, only confirmatory molecular testing using RT-PCR was recommended for suspect cases according to the recommendations of the French government to avoid wastage of reagents. Reference serological testing for IgG antibodies to SARS-CoV-2 was only progressively implemented in France during the study period, to be only available at the end of May, after the beginning of the deconfinement. While the purpose of our study was not to assess the virological analytical performances of the BIOSYNEX® COVID-19 BSS (IgG/IgM) (Biosynex Swiss SA), this rapid diagnostic test has been fully recommended for both SARS-CoV-2-specific IgG and IgM detection by the French Ministry of Health (https://covid-19.sante.gouv.fr/tests. ; last access 25 August 2020), following an official report from the National Reference Center for Respiratory Viruses [Centre National de Référence Virus des infection respiratoires (dont la grippe)], Institut Pasteur, Paris. We have added this information in the text. 

Furthermore, in order to comply with the requirements of the ethical committee, all persons with a positive serological result were referred to the laboratory for diagnostic confirmation and to the hospital for management. In this study, 11 (13.3%) people had a positive result with the self-test and they were oriented to laboratory for result confirmation. We have added these details in the “substudy 2” sections of Methods and results in the revised manuscript.

---

## [Decision Letter · Decision Letter 1]

29 Sep 2020

PONE-D-20-20619R1

Capillary whole-blood IgG-IgM COVID-19 self-test  as a serological screening tool for SARS-CoV-2 infection  adapted to the general public

PLOS ONE

Dear Dr. Belec

Thank you for submitting your manuscript to PLOS ONE. After careful consideration, we feel that it has merit but does not fully meet PLOS ONE’s publication criteria as it currently stands. Therefore, we invite you to submit a revised version of the manuscript that addresses the points raised during the review process.

Please ensure that you address reviewers comments

We look forward to receiving your revised manuscript.

Kind regards,

Alan Landay

Academic Editor

PLOS ONE

Reviewers' comments:

Reviewer's Responses to Questions

**Comments to the Author**

1. If the authors have adequately addressed your comments raised in a previous round of review and you feel that this manuscript is now acceptable for publication, you may indicate that here to bypass the “Comments to the Author” section, enter your conflict of interest statement in the “Confidential to Editor” section, and submit your "Accept" recommendation.

Reviewer #1: (No Response)

Reviewer #2: All comments have been addressed

2. Is the manuscript technically sound, and do the data support the conclusions?

Reviewer #1: Yes

Reviewer #2: (No Response)

3. Has the statistical analysis been performed appropriately and rigorously? 

Reviewer #1: Yes

Reviewer #2: (No Response)

4. Have the authors made all data underlying the findings in their manuscript fully available?

Reviewer #1: Yes

Reviewer #2: Yes

5. Is the manuscript presented in an intelligible fashion and written in standard English?

Reviewer #1: No

Reviewer #2: Yes

6. Review Comments to the Author

Reviewer #1: Further minor corrections are required. There are still language problems.

Note [page 18]: L74: please clarify what ‘overcoming the de-confinement period’ means

Highlight [page 18]: Furthermore, antibody tests for SARS-CoV-2 may constitute one of the keys to fight the SARS-CoV-2 epidemic, in particular to overcome the de-confinement period [9].

Note [page 18]: L76 Delete a priori considered to be healed and protected against new infection

Highlight [page 18]: Seropositivity to SARS-CoV-2 antigens would also allow to identify previously infected individuals, including asymptomatic patients, a priori considered to be healed and protected against new reinfection [9].

Note [page 21]: L140 change is to was

Note [page 26]: L270- insert (Q10) after question

Note [page 26]: Table 1; Previously diagnosed COVID-19 positive among those previously COVID-19 tested. Given the total number is 22+145= 167 shouldn’t the ‘among those previously COVID-19 tested be removed? The previous line of the table indicates only 34 were tested.

Note [page 28]: L296: change 'oriented' to 'referred to a clinically certified laboratory'

Highlight [page 28]: Note that, in this substudy, 11 (13.3%) people had a positive results with the self-test, and they were oriented to laboratory for result confirmation.

Note [page 32]: L380 to 384: better to break this long sentence down into at least two smaller and easier to understand sentences.

Note [page 35]: L 447 change 'pseudo-insurance of immunity or non-contagiousness' to ‘an individual assuming they are immune or non-contagious when they are not. This emphasizes the need for pre- and post-test counseling.’

Highlight [page 35]: The possible risk of adverse effects of the COVID-19 self-test should not be underestimated, such as a pseudo-insurance of immunity or non-contagiousness.

Highlight [page 35]: It will of course be necessary to precisely assess all these potential perverse effects.

Note [page 35]: L450 change 'However,' to 'Nevertheless,'

Note [page 35]: L 449 change sentence’ It will of course..” to “Post-marketing surveillance for these potential adverse consequences will be needed.”

Note [page 36]: L456 delete full

Note [page 37]: l481 change ‘does not eliminate a SARS-CoV-2 infection’ to ‘does not exclude the possibility of active SARS-CoV-2 infection and infectiousness’’

Note [page 37]: L484 change’ accentuating tensions in’ to ‘a burden on’

Note [page 37]: L 485. Change ‘ In any case, the presence of ….”immunological passport” of protection ….immunity (49)’ to “It should be emphasized that it is not known if a positive antibody test represents protection and the concept of an “immunological passport” cannot be supported at this time. (46,47,49)” This change was previously requested and is essential.

Highlight [page 37]: In any case, the presence of reactivities could constitute an "immunological passport" of protection [46,47], although the general assumption is that the presence of antibodies will provide at least some immunity [49].

Note [page 37]: L487: change 'However,…' sentence to “ Even if antibodies to SARS-CoV-2 are shown to offer some level of protection the durability of any such protection is not known at this time. Furthermore the presence of antibodies does not necessarily indicate that that the person is not contagious particularly during the early phases of infection ”

Note [page 37]: L500 Change ‘will’ to ‘may’

Note [page 37]: L 503: delete ‘,lead to partial lifting of “stay-at-home” or “shelter-in-place” orders, and would help get the economy back to normal’

Note [page 38]: L 505 change to ‘UK, has made SARS-CoV-2 home tests available to healthcare workers… ‘

Note [page 38]: L 510: delete’ to individuals and the country’

Note [page 38]: L 510 : change ‘features demonstrate’ to ‘study demonstrates’

Reviewer #2: (No Response)

7. PLOS authors have the option to publish the peer review history of their article (what does this mean?). If published, this will include your full peer review and any attached files.

Reviewer #1: No

Reviewer #2: No

---

## [Author Response · Author response to Decision Letter 1]

1 Oct 2020

Responses to journal requirements and to Reviewers 

Journal Requirements:

Reviewers' comments: 

Reviewer's Responses to Questions

Comments to the Author

1. If the authors have adequately addressed your comments raised in a previous round of review and you feel that this manuscript is now acceptable for publication, you may indicate that here to bypass the “Comments to the Author” section, enter your conflict of interest statement in the “Confidential to Editor” section, and submit your "Accept" recommendation.

Reviewer #1: (No Response)

Reviewer #2: All comments have been addressed

Our answer: We have made minor corrections thorough the manuscript; therefore, we hope that our revised manuscript is presented in an intelligible fashion and written in standard English. 

2. Is the manuscript technically sound, and do the data support the conclusions?

Reviewer #1: Yes

Reviewer #2: (No Response)

3. Has the statistical analysis been performed appropriately and rigorously?

Reviewer #1: Yes

Reviewer #2: (No Response)

4. Have the authors made all data underlying the findings in their manuscript fully available?

Reviewer #1: Yes

Reviewer #2: Yes

Our answer: We thank the reviewers for their nice comments on our work. 

3. Have the authors made all data underlying the findings in their manuscript fully available?

Reviewer #1: Yes

Reviewer #2: Yes

Our answer: We thank the reviewers for their nice comments on our work. 

5. Is the manuscript presented in an intelligible fashion and written in standard English?

Reviewer #1: No

Reviewer #2: Yes

Our answer: In order to acknowledge the comments raised by Reviewer # 1, we have corrected words and grammar as suggested by Referee. 

5. Review Comments to the Author

Answer to reviewer #1

Further minor corrections are required. There are still language problems.

Our answer: In order to acknowledge the comments raised by Reviewer # 1, we have corrected all the confusing sentences raised by the referee. 

Note [page 18]: L74: please clarify what ‘overcoming the de-confinement period’ means

Highlight [page 18]: Furthermore, antibody tests for SARS-CoV-2 may constitute one of the keys to fight the SARS-CoV-2 epidemic, in particular to overcome the de-confinement period [9].

Our answer: We have changed the sentence as following: “Furthermore, antibody tests for SARS-CoV-2 may constitute one of the keys to fight the SARS-CoV-2 epidemic, in particular to overcome the period after lockdown”. 

Note [page 18]: L76 Delete a priori considered to be healed and protected against new infection

Highlight [page 18]: Seropositivity to SARS-CoV-2 antigens would also allow to identify previously infected individuals, including asymptomatic patients, a priori considered to be healed and protected against new reinfection [9].

Our answer: We have simplified this sentence as suggested.

Note [page 21]: L140 change is to was

Our answer: We have changed the words as suggested. 

Note [page 26]: L270- insert (Q10) after question

Our answer: Done

Note [page 26]: Table 1; Previously diagnosed COVID-19 positive among those previously COVID-19 tested. Given the total number is 22+145= 167 shouldn’t the ‘among those previously COVID-19 tested be removed? The previous line of the table indicates only 34 were tested.

Our answer: We thank the reviewer for this pertinent remark. This is a mistake. Indeed, of the 34 people who were previously tested, 22 were found to be positive for SARS-CoV-2 infection. We have corrected this in the revised manuscript.

Note [page 28]: L296: change 'oriented' to 'referred to a clinically certified laboratory'

Highlight [page 28]: Note that, in this substudy, 11 (13.3%) people had a positive results with the self-test, and they were oriented to laboratory for result confirmation.

Our answer: We have changed the words as suggested. 

Note [page 32]: L380 to 384: better to break this long sentence down into at least two smaller and easier to understand sentences.

Our answer: We have changed the sentence as suggested. 

Note [page 35]: L 447 change 'pseudo-insurance of immunity or non-contagiousness' to ‘an individual assuming they are immune or non-contagious when they are not. This emphasizes the need for pre- and post-test counseling.’

Highlight [page 35]: The possible risk of adverse effects of the COVID-19 self-test should not be underestimated, such as a pseudo-insurance of immunity or non-contagiousness.

Our answer: We have changed the sentence as suggested. 

Highlight [page 35]: It will of course be necessary to precisely assess all these potential perverse effects.

Note [page 35]: L450 change 'However,' to 'Nevertheless,'

Note [page 35]: L 449 change sentence’ It will of course..” to “Post-marketing surveillance for these potential adverse consequences will be needed.”

Our answer: We have changed the sentence as suggested. 

Note [page 36]: L456 delete full

Our answer: We have deleted the word as suggested. 

Note [page 37]: l481 change ‘does not eliminate a SARS-CoV-2 infection’ to ‘does not exclude the possibility of active SARS-CoV-2 infection and infectiousness’’

Our answer: We have changed the sentence as suggested. 

Note [page 37]: L484 change’ accentuating tensions in’ to ‘a burden on’

Our answer: We have changed the sentence as suggested. 

Note [page 37]: L 485. Change ‘ In any case, the presence of ….”immunological passport” of protection ….immunity (49)’ to “It should be emphasized that it is not known if a positive antibody test represents protection and the concept of an “immunological passport” cannot be supported at this time. (46,47,49)” This change was previously requested and is essential.

Highlight [page 37]: In any case, the presence of reactivities could constitute an "immunological passport" of protection [46,47], although the general assumption is that the presence of antibodies will provide at least some immunity [49].

Our answer: We have changed the sentence as suggested. 

Note [page 37]: L487: change 'However,…' sentence to “ Even if antibodies to SARS-CoV-2 are shown to offer some level of protection the durability of any such protection is not known at this time. Furthermore the presence of antibodies does not necessarily indicate that the person is not contagious particularly during the early phases of infection ”

Our answer: We have changed the sentence as suggested. 

Note [page 37]: L500 Change ‘will’ to ‘may’

Our answer: Done 

Note [page 37]: L 503: delete ‘,lead to partial lifting of “stay-at-home” or “shelter-in-place” orders, and would help get the economy back to normal’

Our answer: We have deleted the sentence as suggested. 

Note [page 38]: L 505 change to ‘UK, has made SARS-CoV-2 home tests available to healthcare workers… ‘

Our answer: We have changed the sentence as suggested. 

Note [page 38]: L 510: delete’ to individuals and the country’

Our answer: We have deleted the sentence as suggested. 

Note [page 38]: L 510 : change ‘features demonstrate’ to ‘study demonstrates’

Our answer: We have changed the words as suggested.

---

## [Decision Letter · Decision Letter 2]

5 Oct 2020

Capillary whole-blood IgG-IgM COVID-19 self-test  as a serological screening tool for SARS-CoV-2 infection  adapted to the general public

PONE-D-20-20619R2

Dear Dr. Belec

We’re pleased to inform you that your manuscript has been judged scientifically suitable for publication and will be formally accepted for publication once it meets all outstanding technical requirements.

Kind regards,

Alan Landay

Academic Editor

PLOS ONE

Additional Editor Comments (optional):

Reviewers' comments:

Reviewer's Responses to Questions

**Comments to the Author**

1. If the authors have adequately addressed your comments raised in a previous round of review and you feel that this manuscript is now acceptable for publication, you may indicate that here to bypass the “Comments to the Author” section, enter your conflict of interest statement in the “Confidential to Editor” section, and submit your "Accept" recommendation.

Reviewer #1: All comments have been addressed

2. Is the manuscript technically sound, and do the data support the conclusions?

Reviewer #1: Yes

3. Has the statistical analysis been performed appropriately and rigorously? 

Reviewer #1: Yes

4. Have the authors made all data underlying the findings in their manuscript fully available?

Reviewer #1: Yes

5. Is the manuscript presented in an intelligible fashion and written in standard English?

Reviewer #1: Yes

6. Review Comments to the Author

Reviewer #1: (No Response)

7. PLOS authors have the option to publish the peer review history of their article (what does this mean?). If published, this will include your full peer review and any attached files.

Reviewer #1: No

---

## [Editor Report · Acceptance letter]

8 Oct 2020

PONE-D-20-20619R2 

Capillary whole-blood IgG-IgM COVID-19 self-test as a serological screening tool for SARS-CoV-2 infection adapted to the general public 

Dear Dr. Bélec:

I'm pleased to inform you that your manuscript has been deemed suitable for publication in PLOS ONE. Congratulations! Your manuscript is now with our production department. 

Kind regards, 

on behalf of

Prof. Alan Landay 

Academic Editor

PLOS ONE